# Screening for Human T-Cell Lymphotropic Virus (HTLV) in Pregnant Women in the Peruvian Amazon and Systematic Review with Meta-Analysis of HTLV Infection in Peru

**DOI:** 10.3390/pathogens10030260

**Published:** 2021-02-25

**Authors:** José-Manuel Ramos-Rincón, Sonia Ortiz-Martínez, María-Esteyner Vásquez-Chasnamote, Eva de-Miguel-Balsa, Olga-Nohelia Gamboa-Paredes, Michael-John Talledo-Albujar, Giovanni López-Campana, Juan Carlos Celis-Salinas, Laura Prieto-Pérez, Miguel Górgolas-Hernández, Martin Casapía-Morales

**Affiliations:** 1Clinical Medicine Department, Miguel Hernández University of Elche & General University Hospital of Alicante, 03010 Alicante, Spain; 2Medical Practice El Ballestero, Health Service of Castilla La Mancha, 02614 Albacete, Spain; soniaom1978@gmail.com; 3Natural Resources Research Center, Peruvian Amazon National University, Iquitos 16001, Peru; vmariaesteyner.12@gmail.com; 4Clinical Medicine Department, Miguel Hernández University of Elche & General University Hospital of Elche, 03002 Alicante, Spain; eva.miguelb@umh.es; 5Medical Department, Amazon Rainforest Civil Association, Iquitos 16001, Peru; ogamboa@acsaperu.org; 6Virology Unit, Institute of Tropical Medicine ‘Alexander von Humboldt’, Peruvian University Cayetano Heredia, Lima 15102, Peru; michaeltalledo@yahoo.com (M.-J.T.-A.); mariano.lopez.c@upch.pe (G.L.-C.); 7Infectious Diseases and Tropical Medicine Service, Regional Hospital of Loreto & National University of the Peruvian Amazon, Iquitos 1600, Peru; doctorcelis@yahoo.com; 8Infectious Diseases Division, University Hospital Foundation Jiménez Díaz & Autonomous University of Madrid, 28040 Madrid, Spain; lauraprieto83@gmail.com (L.P.-P.); mgorgolas@me.com (M.G.-H.); 9Infectious Diseases and Tropical Medicine Service, Regional Hospital of Loreto & National University of the Peruvian Amazon & Medical Department, Amazon Rainforest Civil Association, Iquitos 16001, Peru; mcasapia@acsaperu.org

**Keywords:** HTLV, prevalence, pregnant women, Peru, Amazon, systematic review

## Abstract

**Background**. Human T-cell lymphotropic virus type 1 (HTLV-1) is responsible for tropical spastic paraparesis and HTLV-1-associated leukemia/lymphoma. The infection is endemic in some areas of Peru, but its prevalence in the Peruvian Amazon is not well established. We aimed to assess the seroprevalence of HTLV-1 infection in pregnant women in the Peruvian Amazon. Moreover, we performed a systematic literature review and meta-analysis of the seroprevalence of HTLV infection in Peru. **(2) Methods**. This is a prospective cross-sectional study involving pregnant women attending health centers in the city of Iquitos, Peru, in May and June 2019. The presence of antibodies against HTLV-1 was assessed using ELISA (HTLV I + II ELISA recombinant v.4.0, Wiener lab, Rosario, Argentina). Positive cases were confirmed by Western Blot and HTLV-1 proviral load. **(3) Results**. The study included 300 pregnant women with a mean age of 26 years (standard deviation [SD] 6.4). Five patients were diagnosed with HTLV-1 infection (prevalence 1.7%, 95% confidence interval (CI) 0.7% to 3.8%). Pregnant women with HTLV-1 infection were discretely younger (mean age 22.6 [SD 22.6] vs. 26.8 [SD 6.3]; *p* = 0.128). None of the five women had been transfused, and all were asymptomatic. Two (40%) also had a positive serology for *Strongyloides*, but larvae were not detected in any of the parasitological stool studies. The systematic review component identified 40 studies, which showed that the prevalence of *HTLV* infection in the general population was 2.9% (95% CI 1.2% to 5.3%) and in women of childbearing age, 2.5% (95% CI 1.2% to 4.0%). (4) Conclusion. The prevalence of HTLV-1 in the Peruvian Amazon basin is about 1.7%, indicating an endemic presence. Screening for HTLV-1 in prenatal care is warranted.

## 1. Introduction

Human T-cell lymphotropic virus (HTLV) was the first human retrovirus identified. It is mostly sexually transmitted [1] but it can also be shed perinatally by breastfeeding [2] and parenterally through contaminated blood, either following transfusions or by needle sharing among injection drug users [3]. Acute infection with HTLV-1 is followed by long-lasting persistence of the virus in all carriers, although clinical manifestations develop only in about 10%. Typically, it presents as a subacute myelopathy known as tropical spastic paraparesis/HTLV-1-associated myelopathy [4] or a lymphoproliferative disorder named adult T-cell leukemia/lymphoma (ATLL) [5].

HTLV can spread through mother-to-child transmission (MTCT), mainly through breastfeeding. As this is one of the main mechanisms of transmission in endemic areas, systematic HTLV antenatal screening is carried out in some countries, such as Japan and a few Latin American countries like Brazil [6,7]. Mothers found to be HTLV positive are advised to substitute breastfeeding with formula feeding for their infants [6].

Worldwide, approximately 5 to 10 million people are infected with the human T-cell lymphotropic virus type 1 (HTLV-1) [8,9]. A large proportion of these people are in Central and South America, where the disease is endemic. In Peru, HTLV-1 prevalence in candidate blood donors varies from 1.2% to 1.7% across regions, with higher figures for some Andean areas [10]. Most studies have taken place in Lima or the Andes [11,12,13], whereas there are scant data from the Amazon [11,14,15,16].

Knowledge on the prevalence of HTLV-1 is vital for understanding whether antenatal screening is justified in the Amazon Basin, so that infected mothers can avoid breastfeeding to effectively prevent vertical HTLV transmission. Thus, this study aimed to assess the seroprevalence of HTLV-1 infection in pregnant women in the Peruvian Amazon. In addition, we undertook a systematic literature review and comprehensive review of the incidence of HTLV-1 in Peruvian cohorts and meta-analyzed the seroprevalence of HTLV infection in Peru.

## 2. Results

### 2.1. HTLV Infection in Pregnant Women

We included 300 pregnant women with an average age of 26 years (standard deviation [SD] 6.4 years). The mean number of deliveries was 2.9 (SD 1.7), and the mean gestation period, 172 days (SD 59). Just under half (44.7%) lived in peri-urban areas, while the rest lived in the city.

Five patients were diagnosed with HTLV-1 infection by serology (prevalence 1.7%, 95% confidence interval [CI] 0.7% to 3.8%). Pregnant women with HTLV-1 infection were slightly younger (mean age 22.6 [SD 22.6] years vs. 26.8 [SD: 6.3] years; *p* = 0.13). None of the five women with HTLV-1 had been transfused, had a tattoo, or had a history of leukemia or neoplasm. All of them with other children had breastfed in the past. *Strongyloides* serology was positive in 2 of the 5 pregnant women (40%); however, the parasitological study in stool was negative for *Strongyloides* larvae in all of them. All HTLV-infected women were asymptomatic.

The characteristics of the five positive cases are summarized in Table 1. The numbers of HTLV-1 pro-viral copies were evaluated in four patients, who presented 3324 copies/10,000 cells, 950 copies/10,000 cells, and <10 copies/10,000 cells; in one, the proviral load was undetectable. No data were available for the remaining patient.

### 2.2. Systematic Review and Meta-Analysis of the Prevalence of HTLV Infection in Peru

The electronic search in PubMed and SciELO yielded 135 records, while a further 6 studies were identified from other sources. After screening the abstracts, we examined the full text of 61 potentially relevant papers, excluding 21 that did not report data on HTLV-1 prevalence. Therefore, the total number of included papers was 40 studies, plus the present study (Figure 1). Figure 1 shows the flow chart for study selection.

Table 2 summarizes the characteristics of the studies from our literature review of HTLV prevalence in Peru [10,11,13,14,15,16,17,18,19,20,21,22,23,24,25,26,27,28,29,30,31,32,33,34,35,36,37,38,39,40,41,42,43,44,45,46,47,48,49,50]. Some studies were performed in the general population cohorts (including prisoners or hospital inpatients), as well as specific populations, like pregnant women or women of childbearing age, sex workers, people with HIV infections, men who have sex with men (MSM), relatives or descendants of people infected with HTLV, people with *Strongyloides* infections, people infected with tuberculosis, and blood donors.

The pooled proportion of HTLV in the general population was 2.34% (95% CI 1.96% to 2.75%; I^2^ 95.51%). Table 3 summarizes the results of pooled prevalence in different groups, assuming a random-effects model. As expected, we found high heterogeneity with the exception of studies related to tuberculosis-infected patients. Figure 2 and Figure 3 summarize the forest plot prevalence in different groups.

## 3. Discussion

Few studies on the prevalence of HTLV have been carried out in pregnant women in Peru. In our study, the prevalence was 1.7%, which is lower than that reported by Wignall et al. [11] and Zurita et al. [22], higher than in the study by Juscamaita et al. [33], and similar to Alarcon et al. [36] a study involving 2492 pregnant women in Lima (the largest sample size in the literature), which also reported a prevalence of 1.7%. The prevalence in pregnant Peruvian women is higher than reported in other South American countries like Argentina (0.19% to 2.25%) [51,52] or Brazil (0.10% to 1.70%) [53,54], and lower than that in studies from French Guyana (4%) [55,56].

HTLV is effectively transmitted from mother to child, mainly through breastfeeding, which is one of the main transmission routes in endemic areas. In fact, pooled prevalence among descendants of HTLV-infected women has been estimated as high as 22.5% in the Peruvian population. However, HTLV screening in women attending antenatal clinics in endemic areas appears cost-effective [57], as preventing MTCT contributes to reducing both the incidence of HTLV and the burden of HTLV-associated diseases [2].

In Peru, HTLV-1 infections are reported in 1.2% to 3.4% of potential blood donors [10,50]. Blood products have been screened for HTLV-1 since 1997 in this country, but there is no specific program in pregnant women or neonates. The prevalence we observed in pregnant women supports the need to implement measures to prevent MTCT. However, to better understand the public health importance of HTLV infections, additional, adequately powered studies examining HTLV seroprevalence across Peru are a priority [2].

The main risk factors for HTLV-1 infection, as described in the literature, are age of more than 30 years, first sexual intercourse before the age of 20 years, and history of abortion or transfusion [30,36,47]. In our study, most infected women had been sexually active before the age of 20 years, but the rest of the factors did not seem to play a role.

Two studies have examined the prevalence of HTLV-1 in the Peruvian Amazon: one in remote indigenous communities, where the prevalence was 1.9% [14], and another in sex workers in Iquitos, who showed a seropositivity of 4.2% [11]. Despite the geographical proximity, the study populations were not comparable. The prevalence in our study was lower than in either of these studies. Nevertheless, in a cross-sectional, population-based study in 14 indigenous communities of the Matsés ethnic group in Requena province (Loreto region), the prevalence was 0.6% lower than that in our study [15].

Incidence of *Strongyloides* infection in patients with HTLV is higher than that in non-infected patients [58]. HTLV infection is considered a risk factor for *S. stercoralis* dissemination, which can be fatal. *S. stercoralis* infection is related to HTLV-1 clonal expansion in asymptomatic individuals [59]. Moreover, in Peru, Gotuzzo et al. reported HTLV infection in 85.7% of *Strongyloides* hyperinfection patients and 9.7% of patients with intestinal *Strongyloides* infection [27]. In our study, *Strongyloides* serology was positive in 40% of pregnant women; however, the parasitological study in stool was negative. Testing for antibodies against *S. stercoralis* is one strategy to identify a *Strongyloides* infection in patients with a negative parasitological study (direct stool smears, the Baermann technique, the Harada-Mori filter paper culture, charcoal cultures, and nutrient agar plate cultures) [60,61]. There is no clear explanation for the high seroprevalence of *Strongyloides* in HTLV-infected women in our study. The Peruvian Amazon may be an endemic region for *Strongyloides*, signaling high prevalence in the general population, including pregnant women [60]. Future studies are needed to evaluate clinical and subclinical *Strongyloides* infection in pregnant, HTLV-infected women.

The main strength of this study is its evaluation of HTLV-1 prevalence in pregnant women in greater Iquitos, providing information that is relevant for monitoring the risk of perinatal transmission through breastfeeding. This study has several limitations, Firstly, it screened a small sample for a prevalence study, so differences in patient characteristics between positive and negative cases could not be detected. Secondly, the proviral load was determined retrospectively, and data could not be collected from one patient who was not followed up. Thirdly, it was not possible to follow up the pregnant women and recommend avoiding breastfeeding for preventing transmission.

We consider necessary a large study in the Peruvian Amazon to understand the relevance of asymptomatic HTLV infections in pregnant women and the coinfection with *Strongyloides*, to provide definitive evidence supporting implementation of screening in pregnant women, as already in place for HIV and other viral infections.

## 4. Materials and Methods

### 4.1. Type of Study and Study Period

We performed a cross-sectional survey of HTLV-1 and strongyloidiasis in May and June 2019.

### 4.2. Study Population and Data Collection

We included pregnant adults (≥18 years) attending health centers in four districts of greater Iquitos, a city in the Peruvian Amazon. Participants were selected through convenience sampling (i.e., on days when the researcher was at the health center) when they visited the midwife for prenatal check-ups. Previous studies performed by our group have reported details of the study population and data collection [60,62].

The presence of antibodies against HTLV-1 was assessed by ELISA (HTLV I+II ELISA recombinant v.4.0, Wiener lab, Rosario, Argentina). Positive cases were confirmed by Western blot and SYBR Green–based real-time quantitative polymerase chain reaction (qPCR) assay, developed in the Institute of Tropical Medicine ‘Alexander von Humboldt’, Peruvian University Cayetano Heredia, Lima [63]. Briefly, DNA from peripheral blood mononuclear cells (PBMCs) was amplified by real-time PCR to detect and quantify HTLV-1 using the Light Cycler 480 II System (Roche Life Sciences). Genomic DNA was extracted from 5 × 10^6^ PBMCs by a spin column using the QIAamp blood DNA mini kit (Qiagen, Hilden, Germany).

Serology was processed for *Strongyloides stercoralis* with the *Strongyloides* IgG IVD-ELISA kit (DRG Instruments GmbH, Marburg, Germany), while Baermann’s method and charcoal culture were performed in stool specimens from HTLV-1 positive patients.

### 4.3. Data Analysis

Patient information was recorded in a Microsoft Excel database. The collected data were analyzed using SPSS statistical software (version 22.0, IBM). We used the chi-squared test or Fisher’s exact test to analyze the presence of HTLV infection according to several demographic variables, considering results to be significant when the *p* value was less than 0.05.

### 4.4. Ethical Considerations

The Ethics Committee of the General University Hospital of Alicante (Spain) approved the project (PI2018/113), as did the Ethics Committee of Loreto Regional Hospital in Iquitos (Peru) (027-CIEI-HRL-2019). After receiving information on the study, individuals who volunteered to participate gave their written consent and were included. All results were kept strictly confidential, and participants with positive results were referred to their health centers for assistance or treatment.

### 4.5. Literature Review of the Prevalence of HTLV Infection in Peru

We performed an electronic search in PubMed and SciELO on 1 April 2020, using the following key words, grouped into three main concepts: “HTLV” AND “Peru”. Studies collected were published from 1 January 1991 to 1 April 2020. The search was limited to studies in humans and written in English, Spanish, Italian, or French. Additional records were identified by means of cited reference searching and electronic searches for gray literature (Google and Google Scholar).

We examined surveys, reports, reviews, and epidemiological studies on the prevalence of HTLV. Titles and abstracts were screened for relevant papers, and full texts were retrieved when appropriate. Data on the prevalence of *HTLV* were extracted, regardless of the type of population (children, adults, immunosuppressed patients, etc.).

We performed a proportion meta-analysis, using the Stuart-Ord (inverse double arcsine square root) method to calculate the 95% coefficient intervals and create the forest plots. Heterogeneity was analyzed using the I^2^ statistic (StatsDirect Statstical software v. 3.3.4) [64]. The prevalence meta-analysis was presented according to six population groups: (1) general population, (2) pregnant women or women of childbearing age, (3) sex workers, (4) people with HIV infections, (5) MSM, (6) relatives or descendants of people infected with HTLV, (7) people with *Strongyloides* infections, (8) people with tuberculosis, and (9) blood donors. We use a random-effects (inverse variance) method.

## 5. Conclusions

The prevalence of HTLV-1 in the Peruvian Amazon basin is about 1.7%. Since HTLV is endemic (prevalence > 1%) in this population of pregnant women, screening for HTLV in prenatal care is warranted.

## Figures and Tables

**Figure 1 pathogens-10-00260-f001:**
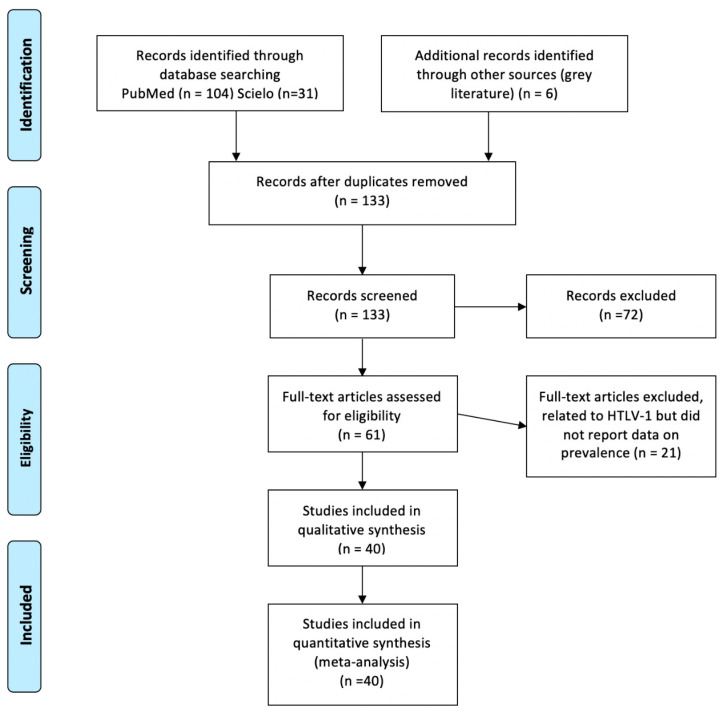
Flow chart for study selection.

**Figure 2 pathogens-10-00260-f002:**
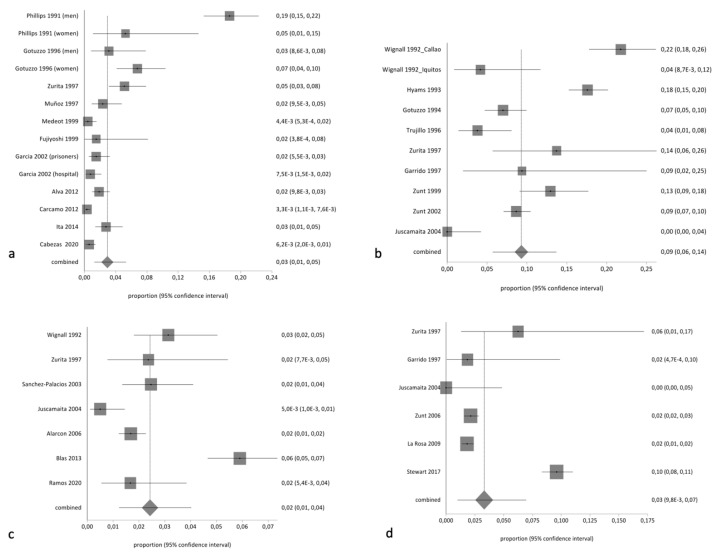
Forest plot of pooled prevalence of HTLV (random effects) in (**a**) the general population, (**b**) sex workers, (**c**) pregnant women or women of childbearing age, and (**d**) men who have sex with men.

**Figure 3 pathogens-10-00260-f003:**
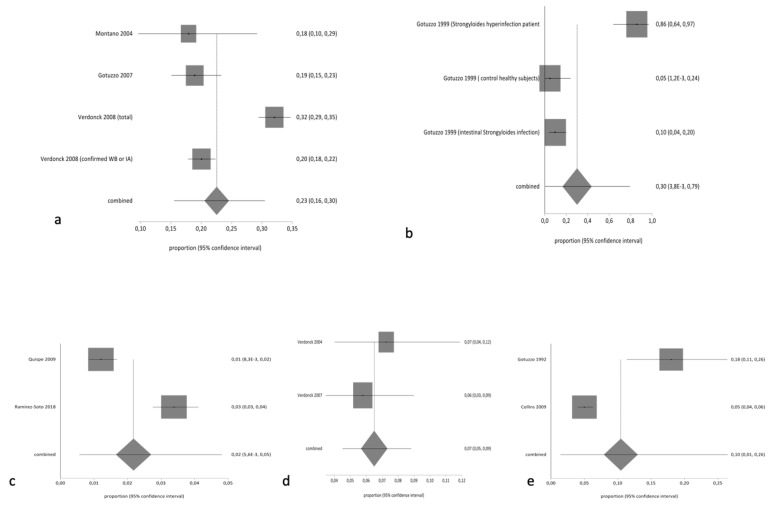
Forest plot of pooled prevalence of HTLV (random effects) in (**a**) descendants and relatives of HTLV patients, (**b**) people with *Strongyloides* infection, (**c**) people living with HIV, (**d**) people infected with tuberculosis, and (**e**) blood donors.

**Table 1 pathogens-10-00260-t001:** Characteristics of five cases of human T-cell lymphotropic virus (HTLV) infection in pregnant women in greater Iquitos, Peru, diagnosed with ELISA serology.

Case	Age, Years	Gestational Age (Days)	Type of Pregnancy	Residence	HTLV-1/2	Proviral Load (Copy /1000 Cell)	*Strongyloides* Serology
1	19	88	Primipara	Urban	HTLV-1	Not available	Positive
2	35	181	Multipara	Urban	HTLV-1	950	Negative
3	20	233	Multipara	Urban	HTLV-1	Undetectable	Negative
4	20	274	Primipara	Periurban	HTLV-1	<10	Negative
5	19	63	Multipara	Periurban	HTLV-1	3324	Positive

**Table 2 pathogens-10-00260-t002:** Peruvian studies of HTLV prevalence.

Study ID	Setting(s)/ (Department)	Population	Prevalence HTLV-1	Diagnostic Procedure
Phillips 1991 [17]	Peru	552 HIV: 495 men and 57 women	18.6% men 5.3% women	-ELISA (Cambridge Bioscience) -Western blot (DuPont) -RIPA
Wignall 1992 [11]	Callao (Lima) Iquitos	395 sex workers in Callao 72 sex workers in Iquitos 510 women attended in antenatal clinic in Lima	21.8% 4.2% 3.1%	-EIA (Cambridge Bioscience; Worchester, MA) -Western blot (DuPont; Wilmington, DE) -RIPA
Gottuzo 1992 [13]	Lima	111 men with HIV	18%	-ELISA (Cambridge Bioscience Worcester, Mass) -Western blot (DuPont, Wilmington, Del)
Hyams 1993 [18]	Callao (Lima)	966 sex workers	17.6%	-ELISA (Cambridge Bioscence, Worcester, MA, USA) -Western blot (DuPont, Wilmington, DE, USA) -RIPA
Gotuzzo 1994 [19]	Lima	400 sex workers	7%	-ELISA (Genetic Systems, Seattle). -Western blot (Cambridge Biotech, Worcester, MA)
Trujillo 1996 [20]	Lima	158 sex workers	3.8%	-ELISA (Cambridge Biotech, Worcester, MA) -Western blot (Cambridge Biotech, Worcester, MA)
Gotuzzo 1996 [21]	Lima	407 (280 women and 127 men) Japanese migrants and their children in Peru	6.8% women 3.2% men	-ELISA (Gentic Systems, Seattle, WA) -Western blot (Cambridge Biotech, Worcester, MA)
Zurita 1997 [22]	Quillabamba and Cuzco	370 volunteers 211 pregnant 51 sex workers 48 MSM 47 associated with ITS	5.1% 2.3% 13.7% 6.2% 8.5%	-ELISA (Genetic Systems, Seattle, WA) -Western blot (Cambridge Biotech, Worcester, MA)
Garrido 1997 [23]	Pisco–Ica	54 MSM 32 Sex workers	1.9% 10.4%	-ELISA (Genetic Systems, Seattle) -Western blot (Cambridge Biotech, Worcester, MA)
Muñoz 1997 [24]	Lima	298 non-intravenous drug users	2.3%	-ELISA (Cambridge Biotech, Worcester, MA) -Western blot (Cambridge Biotech, Worcester, MA)
Medeot 1999 [16]	Peruvian Amazon	456 participants	0.44%	-Particle agglutination (PA) technique (Serodia; Fujirebio, Inc., Tokyo, Japan) -IFA -Western blot (Problot-HTLV-1; Fujirebio, Inc)
Fujiyoshi 1999 [25]	Peru	66 participants from the Aymara ethnic group	1.6%	-Particle agglutination (PA) (Serodia HTLV-1, Fujirebio, Tokyo, Japan) -Western blot (HTLV Blot 2.3/HTLV Blot 2.4; Diagnostic Biotechnology, Singapore)
Zunt 1999 [26]	Callao (Lima)	255 sex workers	13%	-ELISA (Cambridge Bioscience, Worcester, MA) -Western blot (Cambridge Bioscience)
Gotuzzo 1999 [27]	Lima	21 *Strongyloides* hyperinfection patients 21 healthy controls 62 patients with intestinal *Strongyloides* infection	85.7% 4.7% 9.7%	-ELISA (Cambridge Bioscience, Worcester, MA) -Western blot (DuPont, Wilmington, DE)
Garcia 2002 [28]	Lima	400 prisoners (EPCROL) 400 participants from Puente de Piedra Hospital (Lima)	1.5% 0.75%	-ELISA -Western blot
Zunt 2002 [29]	Lima	1119 sex workers	8.7%	-ELISA (Cambridge Bioscience) -Western blot (Cambridge Bioscience)
Sanchez-Palacios 2003 [30]	Huanta El Carmen Lima	568 women	2.5%	-ELISA (Platelia HTLV-I New, Sanofi Diagnostics Pasteur, Paris, France) -IFA (California Department of Health Services, Berkeley, CA, USA) -RIBA (Chiron Corporation, Emeryville, CA, USA) -Western blot (WB, Cambridge Biotech, Worcester, MA, USA) -PCR (Belgium, AM Vandamme)
Verdonck 2004 [31]	Lima	193 tuberculosis patients	7.3%	-ELISA -Western blot
Montano 2004 [32]	Lima	67 children exposed to HTLV-1	18%	-ELISA (Platelia HTLV-1; Sanofi Pasteur) -Western blot (Genelabs Diagnostics)
Juscamaita 2004 [33]	Huamanga (Ayacucho)	602 pregnant women 85 sex workers 74 MSM	0.5% 0% 0%	-ELISA (Vironostika) -Inno-LIA HTLV I/II score (Innogenetics, Ghent, Belgium)
Blas 2005 [34]	Lima	23 Norwegian scabies patients	69.6%	-ELISA (Cambridge Bioscience, Worcester, MA) -Western immunoblot (DuPont, Wilmington, DE)
Laguna-Torres 2005 [35]	Lima y Cuzco	351 multi-transfused patients	3.1%	-ELISA (Vironostika) -lmmunoblot techniques (HTLV blot 2.4 Genelabs Diagnostic)
Alarcon 2006 [36]	Lima	2492 pregnant women	1.7%	-ELISA (Cambridge Bioscience Corp, Worcester, MA) -Western blot (Cambridge Bioscience Corp)
Zunt 2006 [37]	6 Peruvian cities (Arequipa, Iquitos, Pucallpa, Lima, Piura and Sullana)	2073 MSM	2.1%	-ELISA (Vironostika) -Western blot (HTLV-I/II blot 2.4; Genelabs Diagnostics, Singapore)
Zunt 2006 [38]	Lima	53 myelopathy patients	81%	-ELISA (Cambridge Bioscience, Worcester, MA) -Western blot (Genelabs Diagnostics, Singapore)
Gotuzzo 2007 [39]	Lima	370 descendants of mothers infected with HTLV-1	19% (there were 641 descendants, but only 370 agreed to be tested)	-ELISA (Sanofi Diagnostics Pasteur, France; Bio-Rad Laboratories, U.S.A.; or Cambridge Biotech, USA) -Western blot (Genelabs Diagnostics, Singapore) or (Innogenetics, Belgium)
Verdonck 2007 [40]	Lima	311 tuberculosis patients	5.8%	-ELISA (Ortho y Platelia) -Inno-LIA
Verdonck 2008 [41]	Lima	1233 relatives of people infected with HTLV-1	32% 20% (confirmed WB or immunoassay)	-ELISA (Sanofi Diagnostics Pasteur, Marnes-la-Coquette, France; Bio-Rad Laboratories, Hercules, CA, USA; or Cambridge Biotech, Worcester, MA, USA) -Western blot (Genelabs Diagnostics, Singapore)
Quispe 2009 [10]	Arequipa	2732 blood donors	1.2%	-ELISA (Vironostika HTLV-I/II, BioMerieux, France) -BioELISA HTLV-I+II (BioKit S.A., Spain) -Western blot (WB 2.4, Genelabs Diagnostics, Singapore)
La Rosa 2009 [42]	Arequipa, Iquitos, Pucallpa, Lima, Piura, Sullana	2655 MSM	1.8%	-ELISA (Vironostika) -Western blot (HTLV-I/II blot 2.4; Genelabs Diagnostics, Singapore)
Collins 2009 [43]	Lima	1456 HIV patients	5%	-EIA (Sanofi Pasteur, France or Abbott, USA) -Western blot (Cambridge Biotech, USA or DAVIH-Blot, Cuba)
Delgado 2011 [44]	Lima	44 herpes zoster patients	5%	-ELISA (Ortho)
Alva 2012 [14]	10 indigenous groups in the Amazon less than 12 h by boat from 4 ports: Iquitos, Pucallpa, Yurimaguas y Puerto Maldonado	638 participants	1.9%	-ELISA (Vironostika, NC) -Western blot (HTLV-1/2 Blot 2.4; Genelabs Diagnostics, Singapore)
Carcamo 2012 [45]	24 cities with more than 50,000 inhabitants. Branch of the PREVENT trial	1530 serum samples from participants aged 18 to 29 years old	0.3%	-ELISA (Vironostika, Durham, NC, USA) -Western blot 2·4 (Genelabs Diagnostics, Science Park, Singapore)
Mayer 2013 [46]	Lima	893 adults (428 close relatives of a person with HTLV-1 infection, 118 with neurological disorder, 160 with strongyloidiasis, and 181 others)	26.4%	-ELISA (Murex, Ortho y Platelia) -Western blot -Inno-LIA
Blas 2013 [47]	Lima and Pucallpa	1253 women of the Shipibo-Konibo ethnic group	5.9%	-HTLV ELISA (Vironostika, North Carolina) -Western blot (HTLV-1/2 Blot 2.4, Genelabs Diagnostics, Singapore)
Ita 2014 [48]	Provinces of Cangallo, Vilcashuaman and Parinacochas (Ayacucho)	397 participants	2.8%	-ELISA HTLV I+II Murex Biotech Ltd., Dartford, UK. -HTLV-I/HTLV-II Ab.capture ELISA test system (Ortho Clinical Diagnostic, Amersham, UK) -Inno-LIA HTLV-I/II Score (Innogenetics, Ghent, Belgium) -Western blot HTLV Blot 2.4 (MP Diagnostics, Singapore)
Stewart 2017 [49]	Callao (Lima)	1918 sex workers (n = 1477 in period 1, n = 62 in period 2, and n = 379 in period 3)	9.6%	-HTLV EIA (Cambridge Bioscence, Worcester, MA) -Western blot (Cambridge Bioscence) -Vironostika HTLV I/II Microelisa System, Organon Teknika/Biomérieux, Durham, NC -HTLV I/II Western blots (Genelabs Diagnostics, Singapore) -Bioelisa HTLV I/II 5.0, Biokit, Barcelona, Spain -HTLV I/II score; Innogenetics, Ghent, Belgium
Ramirez-Soto 2018 [50]	Abancay	2895 blood samples	3.4%	-ELISA (Biokit, Barcelona, Spain)
Cabezas 2020 [15]	Requena (Loreto)	806 participants (adults and children)	0.6%	-ELISA (bioELISA) -LIA (Inmunogenetic)
Ramos et al. (present study)	Iquitos	300 pregnant women	1.7%	-ELISA (HTLV I+II ELISA recombinante v.4.0, Wiener lab, Rosario, Argentina)

HTLV: human T-cell lymphotropic virus; STI: sexual transmitted infection, MSM: men who have sex with men; PCR: polymerase chain reaction; WB: Western blot.

**Table 3 pathogens-10-00260-t003:** Pooled prevalence of HTLV infection in different population groups.

Population Group	Pooled Prevalence	95% CI	I^2^%
Pregnant women	0.025	0.012–0.040	90.7
Sex workers	0.093	0.056–0.137	92.9
HIV infection	0.104	0.013–0.64	94.8
Men who have sex with men	0.033	0.009–0.069	97.1
Relatives/descendants of HTLV patients	0.225	0.155–0.304	94.7
*Strongyloides* infections	0.300	0.003–0.792	96
Tuberculosis	0.065	0.045–0.088	0
Blood donors	0.021	0.005–0.048	96.8
General population	0.029	0.012–0.053	95.4

Subgroup differences: total Chi² = 2.44 |Chi| = 49.68 (8 degrees of freedom) *p* < 0.001.

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
