# Peer review of "Screening for Human T-Cell Lymphotropic Virus (HTLV) in Pregnant Women in the Peruvian Amazon and Systematic Review with Meta-Analysis of HTLV Infection in Peru"

_pathogens, 2021, doi:10.3390/pathogens10030260_

Round 1
Reviewer 1 Report
This revised resubmission is significantly improved and provides a thorough assessment of the impact of HTLV-1-infections in pregnant women in Peru, and makes a strong case for the need for antenatal testing and reporting for pregnant patients. Using a combination of primary clinical diagnostic studies and a comprehensive literature review, this work gives an accurate account of HTLV-1-infections in Peru as well as in the surrounding countries of the Amazonian Basin. This is a nice study and will serve as a comprehensive resource for healthcare professionals in the region. The authors have adequately addressed all of the concerns of this reviewer.
Author Response
Response:
Reviewer 1. Thanks for the words of the reviewer.
Reviewer 2 Report
The resubmitted manuscript describes a cross-sectional study to provide information on HTLV-1 prevalence in pregnant women in the endemic area of Peruvian Amazon. In addition, the manuscript provides information on Strongyloides co-infection in HTLV-1 positive patients. Moreover, the manuscript also contains a systematic review of HTLV infection in Peru, with meta-analysis, that furnishes a complete information on existing data concerning HTLV-1 infection in Peru. In the discussion, authors admit the main limits of the study, such as the relatively small cohort of women enrolled and absence of follow up to verify incidence of the survey for preventing maternal transmission and avoiding breastfeeding.
Results and discussion are clear, however some minor points need further clarification.
- Abstract. The study, even in the resubmitted version, is described as a “prospective cross-sectional study”. In the opinion of this reviewer the term “prospective” implies a study with follow-up visits and possibly with multiple tests to see if the enrolled individuals develop a condition. This is not the case for this study. In fact, in the “Material and Methods” section, the study is correctly classified simply as “a cross-sectional survey of HTLV-1”. The term “prospective” in the abstract must be cancelled.
- The sentence “In figure 2 and 3 summarizes the forest of forest plot prevalence in different groups” that describes figures 2 and 3, seems not enough clear to this reviewer and should be rephrased.
- References and data reported in Figures 2 and 3 are illegible. Authors should ameliorate the graphics.
Author Response
Reviewer 2.
1.Abstract. The study, even in the resubmitted version, is described as a “prospective cross-sectional study”. In the opinion of this reviewer the term “prospective” implies a study with follow-up visits and possibly with multiple tests to see if the. enrolled individuals develop a condition. This is not the case for this study. In fact, in the “Material and Methods” section, the study is correctly classified simply as “a cross-sectional survey of HTLV-1”. The term “prospective” in the abstract must be cancelled.
Response: According the reviewer the term “prospective” has been cancelled of the manuscript.
2. The sentence “In figure 2 and 3 summarizes the forest of forest plot prevalence in different groups” that describes figures 2 and 3, seems not enough clear to this reviewer and should be rephrased.
Response: Thanks to the reviewer, we have rephrased the sentence.. In figure 2 and 3 summarizes the forest of forest plot prevalence in different groups. So the new phase is In figure 2 and 3 summarizes the forest plot of prevalence of HTLV in the general population, sex workers, pregnant women or women of childbearing age, and men who have sex with men (Figure 2) and descendants and relatives of HTLV patients, people with Strongyloides infection, people living with HIV, people infected with tuberculosis, and blood donors (Figure 3)
3. References and data reported in Figures 2 and 3 are illegible. Authors should ameliorate the graphics
Response: We have ameliorated the graphics